# Sub-nanowatt resolution direct calorimetry for probing real-time metabolic activity of individual *C. elegans* worms

Sunghoon Hur [1,5], Rohith Mittapally [1,5], Swathi Yadlapalli [2✉], Pramod Reddy [1,3✉] & Edgar Meyhofer [1,4✉]

Calorimetry has been widely used in metabolic studies, but direct measurements from individual small biological model organisms such as *C. elegans* or isolated single cells have been limited by poor sensitivity of existing techniques and difficulties in resolving very small heat outputs. Here, by careful thermal engineering, we developed a robust, highly sensitive and bio-compatible calorimetric platform that features a resolution of ~270 pW—more than a 500-fold improvement over the most sensitive calorimeter previously used for measuring the metabolic heat output of *C. elegans*. Using this calorimeter, we demonstrate time-resolved metabolic measurements of single *C. elegans* worms from larval to adult stages. Further, we show that the metabolic output is significantly lower in long-lived *C. elegans daf-2* mutants. These demonstrations clearly highlight the broad potential of this tool for studying the role of metabolism in disease, development and aging of small model organisms and single cells.

[1] Department of Mechanical Engineering, University of Michigan, Ann Arbor, MI 48109, USA. [2] Department of Cell and Developmental Biology, University of Michigan, Ann Arbor, MI 48109, USA. [3] Department of Materials Science and Engineering, University of Michigan, Ann Arbor, MI 48109, USA. [4] Department of Biomedical Engineering, University of Michigan, Ann Arbor, MI 48109, USA. [5]These authors contributed equally: Sunghoon Hur, Rohith Mittapally. ✉email: swathi@umich.edu; pramodr@umich.edu; meyhofer@umich.edu

Calorimetry, a technique that can be utilized to quantify the amount of heat released or absorbed during chemical or physical processes, has been instrumental in analyzing the thermodynamics of reactions[1,2]. Importantly, calorimeters are also critical for analyzing the metabolism of living cells and organisms, which is defined as the sum total of all the heat output associated with the bio-chemical processes in cells[3]. Because of the central role that energy and metabolism play in the normal function of cells[4], it is not surprising that metabolic changes are implicated in aging[5,6] and human diseases, including cancers[7]. Calorimetric measurements, which directly record the heat generated by a (biological) system, represent a non-invasive approach for accurate quantification of metabolic activity, as the metabolic heat output is an integrated signal of both aerobic and anaerobic processes[8–10]. However, such calorimetric measurements have not been able to resolve the heat output from small individual organisms, such as the nematode worm *Caenorhabditis elegans* (*C. elegans*), or isolated cells, owing to poor sensitivity, insufficient long-term stability, and challenges in physiological compatibility.

In this work, we describe a tool capable of systematically studying the metabolism of individual *C. elegans*, which is a widely used model organism because of its ease of maintenance and short generation times[11,12]. We note that this tool is expected to be broadly useful, as *C. elegans* metabolism is being actively investigated owing to its promise for providing insights into human disease and ageing[5,6,10,13–16]. For example, it is well known that genetic mutations, such as *age-1* and *daf-2* mutants, increase the lifespan of the worms mediated via various transcription factors with roles in insulin signaling, autophagy, and cellular energy metabolism[16–18]. Thus, metabolic heat output measurements on single worms through direct calorimetry can provide fundamental insights into metabolic pathway regulation in the context of the biological mechanisms mentioned above. We note that so far *C. elegans* metabolic heat output studies have been limited to large populations of worms as existing calorimeters lack the desired sensitivity (sub-nW resolution) to resolve signals from a single worm[8–10,19]. In fact, the highest resolution claimed for biological calorimetry is at the few nanowatt level (1.9 nW)[20] and was achieved by miniaturizing calorimetric platforms[20,21] to such small sizes that they are unsuitable for studying relatively larger biological samples such as *C. elegans*. As a result, the most-sensitive calorimeter used for measuring *C. elegans* was limited to a resolution of 170 nW[19], making it impossible to study metabolism from individual worms. Finally, we note that none of the available tools have been able to monitor real-time metabolic heat changes with respect to worm size, their activity levels and individual physiological or genetic differences, posing significant barriers to progress.

## Results

**Design of the calorimeter.** Our calorimeter provides a heat resolution of ~270 pW—a 500-fold[19] improvement over calorimeters employed for past *C. elegans* studies and a 10-fold[20,21] improvement compared with the state-of-the-art bio-calorimeters. Further, our approach also incorporates a fluidic environment and optical imaging capabilities. Our instrument is made possible by integrating three subsystems (see Fig. 1a): first, a thermal system composed of three nested shields (Fig. 1a, b) made from copper (whose temperature is feedback controlled) that house two capillary tubes—a sensing capillary tube containing the sample of interest and a matching capillary tube, which acts as a reference (explained in more detail below)—ensures that ambient temperature fluctuations have a minimal effect on the temperature of the capillary tubes. Second, a fluidic system enables transfer of individual worms into the sensing capillary tube and maintaining them under physiological conditions (Fig. 1a). Third, an inverted microscope-based imaging system facilitates tracking of the activity of the biological specimen (Fig. 1a).

To elaborate, the shields are comprised of an outer shield (OS), a middle shield (MS), and an inner shield (IS) (Fig. 1b), which are nested into each other and are mechanically held together by thermally poorly conducting mechanical contacts, which we refer to as weak thermal links, to provide excellent thermal isolation from the ambient. The OS also serves as a vacuum chamber such that all the components included in it (including the capillary tubes) are maintained at a vacuum level below 10 μTorr. Further, the OS features custom-made instrumentation, fluidic and optical feedthroughs, and is actively controlled to a temperature stability of ±1 mK (all temperature signals are measured in a bandwidth of 0.1 Hz, unless mentioned). The MS and IS provide temperature stabilities down to ±15 μK (Supplementary Fig. 6) through active proportional-integral-derivative (PID) feedback control (see Methods). We note that this temperature stability (with direct optical access) is comparable to the best stabilities achieved previously in similar approaches, but without optical access[22,23]. The sensing and matching capillary tube systems are suspended on the IS (Fig. 1c). Both tubes are instrumented with thermistors at the center of the tubes (Fig. 1c). The two capillary tubes have nominally identical dimensions and are 20 mm-long borosilicate tubes with a square cross-section, (250 μm × 250 μm outer dimensions and 50 μm wall thickness). Further, the tubes are coated with a 100 nm-thick layer of gold on all sides except a small portion in the center that provides optical access for imaging (Fig. 1d). A 125 μm-diameter borosilicate rod is inserted from one end into the sensing capillary tube until it reaches the center and serves to localize the worm in the center of the capillary tube where the thermistor is mounted (Fig. 1d). The entire system is mounted on an inverted microscope and the sensing capillary is imaged through a 10x objective to monitor the specimen's activity. Finally, the fluidic system consists of a syringe pump at one end, for manipulating the specimen and flow rate, and a reservoir (petri dish) at the other end, for loading or unloading specimens (Fig. 1a).

**Characterization of thermal resolution.** The principle of operation of our calorimeter can be understood by noting that when heat is generated inside the sensing capillary tube (e.g., owing to the metabolic output of a sample) its temperature rise, ($\Delta T_{th}$), is detected by the thermistor and can be directly related to the metabolic heat output ($\dot{Q}_{metabolic}$) via $\dot{Q}_{metabolic} = G_{th} \times \Delta T_{th}$, where $G_{th}$ is the thermal conductance of the sensing capillary tube and can be directly measured (see Methods). Thus, the heat output resolution is determined by the thermal conductance of the capillary tube (low values of $G_{th}$ are desirable for increased resolution of $\dot{Q}_{metabolic}$) and the temperature resolution of the thermistor[24,25]. In our system, we achieve a small $G_{th}$ by carefully engineering the capillary tube to minimize the contribution of the various heat pathways (conduction, convention and radiation). Specifically, the conductive contribution to $G_{th}$ is minimized by choosing a sufficiently small cross-section ($G_{capillary} = 9.6$ μW K$^{-1}$, $G_{buffer} = 6$ μW K$^{-1}$), whereas the thin gold coating ($G_{Au} = 2.7$ μW K$^{-1}$) on the capillary reduces the radiative contribution to $G_{th}$. Nevertheless, providing optical access to the calorimeter results in some unavoidable radiative coupling ($G_{rad} \sim 6$ μW K$^{-1}$) to the outer shield. As the capillary is operated in a vacuum system, the remaining convective and conductive contributions are reduced to negligible levels (cf. $G_{th,air} \sim 250$ μW K$^{-1}$). In total, the $G_{th}$ under operating conditions (flow rate of 100 nl min$^{-1}$) is 27 μW K$^{-1}$ (see Fig. 2a, Supplementary Fig. 1), and the noise floor

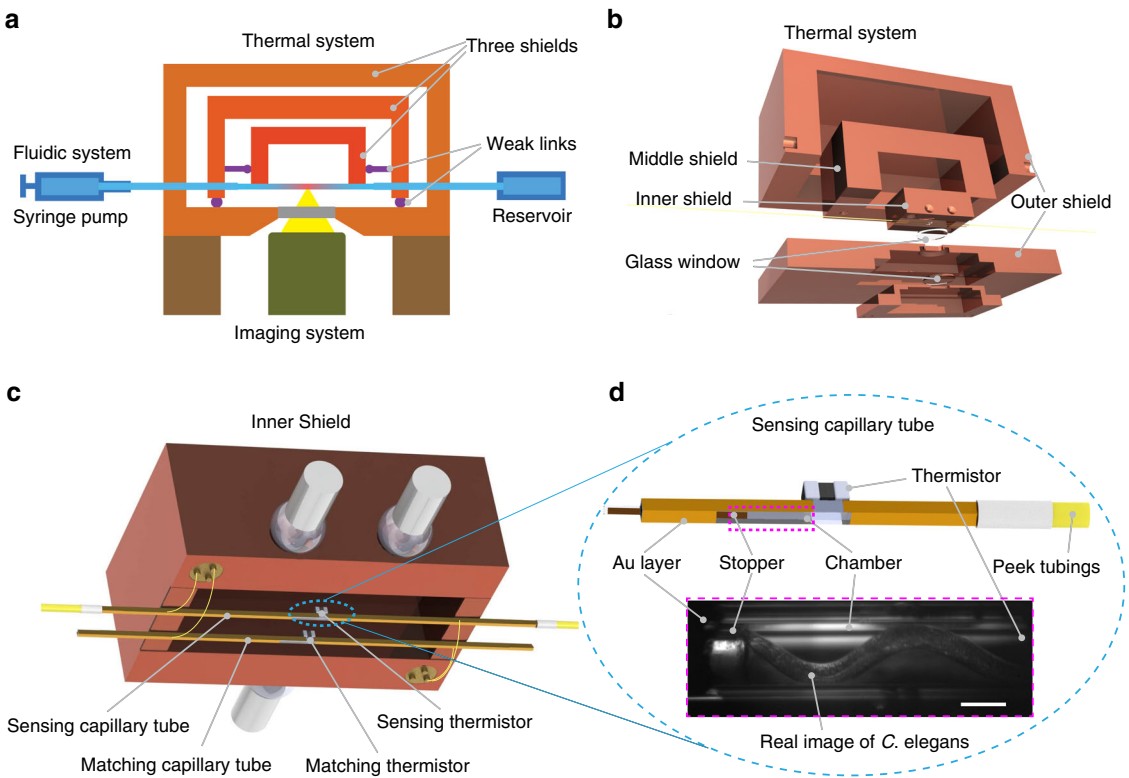

**Fig. 1 Experimental setup. a** Schematic depicting the three subsystems in our calorimeter: the fluidic system (shaded in blue) with the syringe pump and reservoir for sample handling, the optical imaging system (green) focused to the center of the capillary tube and the thermal system involving the three shields with weak thermal links (between OS and MS, MS and IS shown in purple) mounted on the microscope stage. **b** Detailed isometric expanded view of the thermal system showing the assembly of all components. **c** Illustration of IS assembly (not to scale) showing two capillary tubes, the sensing capillary with a PEEK tubing assembly and the matching capillary, with respective thermistors connected in a Wheatstone bridge configuration to extract differential thermal signal. **d** Magnified view of the marked area in **c**, depicting the capillary tube coated with gold on all four sides, except for the central region, where the *C. elegans* is localized, and instrumented with a thermistor to record the temperature. A borosilicate stopper traps the *C. elegans* in the center of the sensing capillary tube. A CCD image (marked in magenta dashed line) of a trapped *C. elegans* during a reference measurement shows the corresponding stopper and Au layer. Images were reproducibly taken every second during the measurement. Scale bar: 100 μm.

of the temperature measurement is 10 μK (see Fig. 2a and inset). As a result, our calorimetry successfully detects 270 pW of heat output ($\dot{Q}_{\text{metabolic}} = G_{\text{th}} \times \Delta T_{\text{th}} = 27\,\mu\text{W K}^{-1} \times 10\,\mu\text{K}$).

We achieved excellent temperature resolution ($\Delta T_{\text{th}} = 10\,\mu\text{K}$), as reflected in the precision of the temperature measurements shown in Fig. 2a and its inset (see Methods section titled Thermal conductance measurement and power resolution for more details). We accomplished this high-temperature resolution by employing an AC-excited Wheatstone bridge (see Methods) in an environment where thermal fluctuations of the capillary tube were attenuated via feedback-controlled shields (described above). Further, we took advantage of a thermistor embedded into the matching capillary tube (see Fig. 1c) to attenuate common-mode temperature drifts. As shown in Fig. 2c and its inset, the temperature of the sensing capillary ($T_{\text{sen}}$) follows that of the matching capillary ($T_{\text{mat}}$), indicating a common-mode radiative coupling to the environment, whereas the differential signal ($T_{\text{th}}$) can be observed to be stable to within ±5 μK. Taken together, this approach enables us to accomplish a noise equivalent temperature (NET) resolution of ±5 μK in 2 h and ~10 μK drift (bandwidth of 5 mHz) in 24 h (Fig. 2c). This implies that, in addition to being able to resolve transient heat output changes (occurring in mins) of 270 pW (Fig. 2b), our system can track heat output changes over a day with same accuracy. Thus, our instrument is well-suited for real-time metabolic measurements on several small biological specimens like *C. elegans*. The

temporal resolution of our calorimeter is set by the thermal time constant of the device which is ~100 s (Fig. 2d).

**Heat output measurement protocol and analysis**. To establish the applicability of our system to biological measurements, we performed metabolic measurements on individual *C. elegans* at 25 °C. A typical measurement in our calorimeter involves five phases (Fig. 3a): first, establishing a baseline $T_{\text{th}}$ (shown in gray) without a worm in the sensing capillary and a flow rate of 100 nl min⁻¹ (all measurements reported here were performed at this flow rate). The worm is then loaded from an external reservoir under visual inspection at a larger flow rate of ~2–10 μl min⁻¹ (shown in violet). Once the worm is loaded and properly localized in the sensing capillary tube, the flow rate is reduced to 100 nl min⁻¹ for continuously recording its heat output and for monitoring its activity (see Methods) via video microscopy (indicated by orange region in Fig. 3a) for a desired time (~1–2 h). This optimal flow rate was chosen so as to both replenish the oxygen content in the fluidic medium (typical oxygen concentration < 8 mg O₂ l⁻¹ at atmospheric pressure), thus avoiding oxygen deprivation of the worm (maximum reported[8] oxygen consumption of 16 ng O₂ h⁻¹ per worm), and holding the worm in the desired location, which is facilitated by both the flow and the borosilicate fiber stopper (see Fig. 1d). This approach solves the difficult task of constraining the worm[26] to the center of the sensing capillary, whereas not inhibiting its natural swimming behavior. After monitoring the worm for the

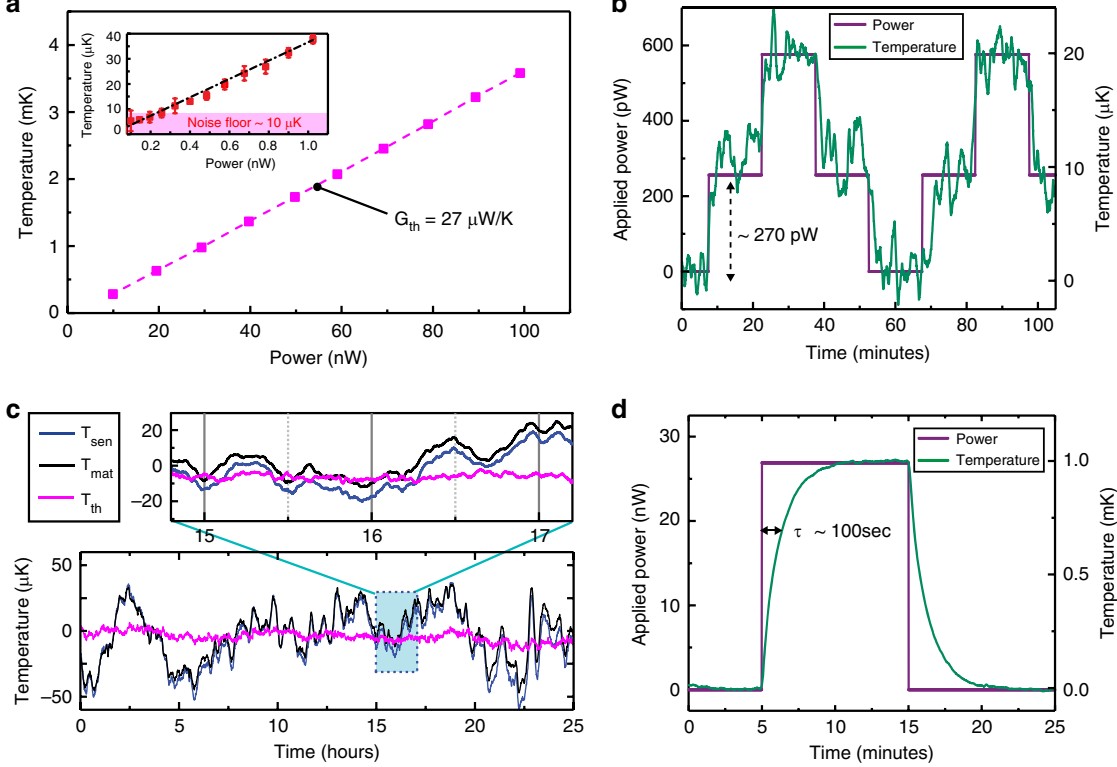

**Fig. 2 Thermal characterization of the calorimeter. a** Thermal conductance characterization ($G_{th}$) performed at a flow rate of 100 nl min$^{-1}$: temperature rise ($\Delta T_{th}$) of the center of the sensing capillary tube as a function of the DC power input is plotted, resulting in $G_{th}$ of ~27 μW K$^{-1}$ (magenta squares). The inset shows the temperature rise for sub-nW level inputs that result in a smaller temperature rise (0–40 μK). The data represent the mean value of four different measurements and the error bars represent the standard deviation (±SD). From the observed standard deviation of (±5 μK) we estimate a noise floor of ~10 μK. **b** Validation of thermal resolution. A thermal pulse (purple solid line) with steps of 270 pW applied on the capillary (see Methods) resulted in a corresponding 10 μK rise on the temperature signal (green solid line), validating our calorimeter's capability to resolve at least 270 pW (heat resolution). **c** Temperatures of both the capillaries ($T_{mat}$ in solid black, $T_{sen}$ in solid blue) without any sample in the sensing capillary tube are plotted for a 25-hour timespan, indicating fluctuations of ±5 μK in the differential signal ($T_{th}$ in solid magenta). A 2 hour window (blue box) clearly shows that $T_{mat}$ follows $T_{sen}$, indicating a common-mode radiative coupling to the outer shield. **d** Time constant of the sensing capillary tube. A square thermal pulse (purple solid line) in steps of ~27 nW results in a first order temperature response with a time constant of ~1.5 min.

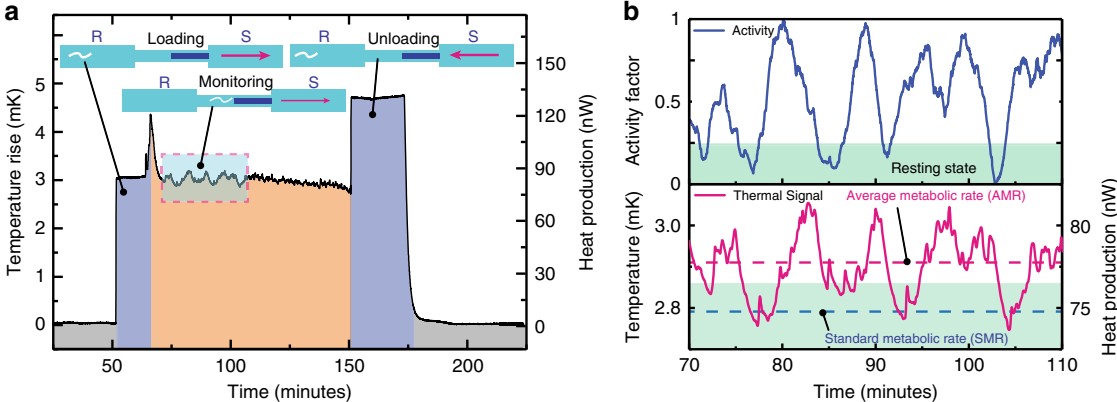

**Fig. 3 C. elegans metabolic heat output measurement and analysis. a** A typical procedure to measure heat output from a single *C. elegans* involves establishing an initial reference (gray region), followed by loading the worm (violet region) from the reservoir side (R) with a syringe pump (S) at withdraw flow rates (R → S) of ~2–10 μl min$^{-1}$. The orange region indicates the monitoring phase (1–2 h), followed by unloading (R ← S, violet region) at flow rates of ~2–10 μl min$^{-1}$. **b** Temperature rise (magenta plot in bottom panel) and activity factor (blue plot in top panel) of a reference measurement (marked blue region in Fig. 3a). In the panel showing the activity factor, the worm's resting activity level is highlighted in green. From the corresponding metabolic signal, SMR (blue dashed line, bottom panel) is derived. AMR (magenta dashed line, bottom panel) is determined by averaging the thermal signal in a 30-minute time interval.

desired time interval, it is unloaded by reversing the flow at rates of 2–10 µl min⁻¹ (shown in violet in Fig. 3a). Once the worm is unloaded from the sensing capillary, the temperature (shown in gray) of the capillary is again recorded to confirm stability of the reference temperature ($T_{th}$). The worm is then transferred to a nematode growth medium (NGM) Agar plate, to feed and grow (room temperature 22 °C), and the measurement cycle is repeated the following day, thus monitoring the metabolic heat changes of the same individual worm through its life cycle. Maintaining such a small heat resolution while continuously replenishing nutrients and oxygen has not been possible in the most-sensitive calorimeter[20], where the cells die within 20 min of their measurement owing to lack of oxygen.

In all our measurements, we observe that the worm's activity with prolonged periods of rapid swimming and resting periods resulted, as expected, in distinct thermal signals. A normalized activity factor (see Methods) that quantifies worm activity shows that the metabolic heat output is strongly correlated with activity (correlation coefficient ~0.82, see Fig. 3b). Note that the blue line in Fig. 3b shows a representative activity trace from a single *C. elegans*, whereas the magenta line presents simultaneously obtained metabolic heat output (see Supplementary Movie 1 depicting the variation of thermal signal and activity with real-time imaging). From the measured thermal signal we extract two metabolic heat output readings, which we call the average metabolic rate (AMR) and the standard metabolic rate (SMR) (magenta and blue dashed lines, respectively, in Fig. 3b) at 25 °C. AMR is the average of the thermal signal in a 30-min window (after an initial thermal equilibration time of 15 min) and SMR is the average of the metabolic heat output in the worm's resting state. We define the worm to be in resting state when activity factor is below 0.25 (green shaded region in Fig. 3b). These measurements clearly demonstrate our system's unique capability to monitor activity-related metabolic changes in single nematode worms.

**Metabolic measurements on N2 wild-type at different stages.** To investigate the metabolic changes during their life-cycle, we monitored the AMR of N2 wild-type strain from L1 larval stage to

the adult stage (day 4). Since the size of *C. elegans* increases by more than an order of magnitude over the course of their lifetime, it is expected that early stage *C. elegans* metabolic heat output is smaller. As plotted in Fig. 4a, our measurements reveal the metabolic heat output to vary from ~4 nW for the L1 stage to ~100 nW for the adult stage, more than a 20-fold change. The metabolic heat output increased as the worm progresses from L1 larval stage to the adult stage (day 4), and is consistent with the previously observed oxygen consumption rates and heat output measurement performed on large number of worms[10,14,27]. Furthermore, we report metabolic heat output measurements from 61 N2 wild-type specimens (Fig. 4b, red triangles for AMR and blue triangles for SMR) as a function of mass. The mass (volume times density) of each worm was estimated by analyzing the 2D images captured from the imaging system (see Methods). These measurements indicate that the average mass-specific AMR and SMR (inset of Fig. 4b) are 55.8 ± 14.45 µW mg⁻¹ and 52.17 ± 12.65 µW mg⁻¹, respectively, demonstrating our calorimeter's ability to delineate small changes in activity-related metabolic heat output from that of the resting state (~7% increase above SMR). In these measurements, the metabolic heat output of individual *C. elegans* worms increased with activity from 5 to 25% relative to the observed SMR. We note that a previous study[28] reported a 50% difference in AMR and SMR as opposed to the 7% observed in our study, but a direct comparison of the results is challenging owing to important differences in the experimental conditions. Specifically, we acquired the SMR of worms without restraining their activity while monitoring the metabolic rate, whereas the earlier report[28] immobilized *C. elegans* either by levamisole-induced paralysis or by genetic paralysis. It is known that the relationship between any organism's metabolic rate and size is expected to follow a simple exponential scaling law[29], described as $Q = a \cdot M^b$, where $Q$, $M$, $a$, and $b$ are standard metabolic heat output, body size, scalar constant, and exponential constant, respectively. Most studies focused on this exponential constant $b$ across species (inter-specific) of animals ranging from unicellular microbes to elephants[30], including in *C. elegans*[31]. Here, we deduce an exponential constant of 0.76 ± 0.06 (Fig. 4c) from the SMR of wild-type worms (intra-specific) at different

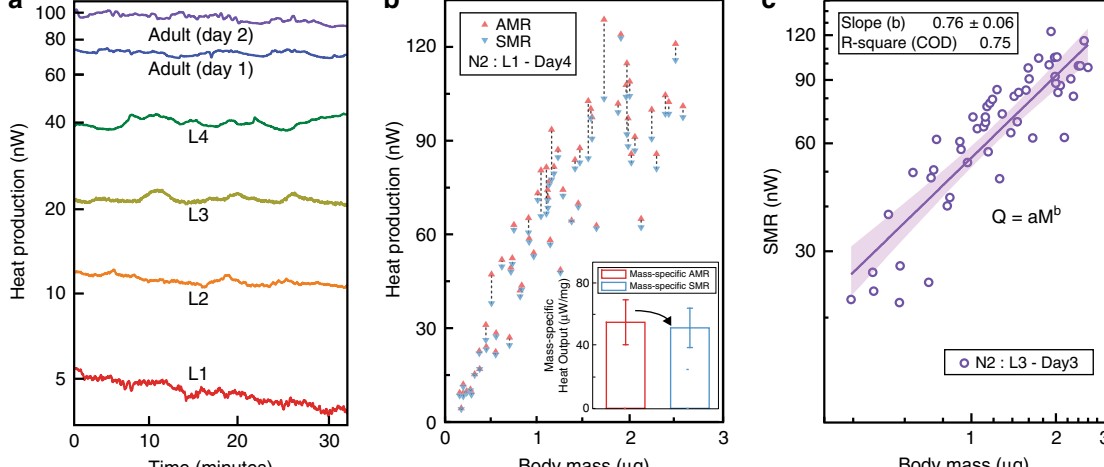

**Fig. 4 Size-dependent metabolic heat output measurements on N2 wild-type. a** Real-time metabolic heat output profiles of referential *C. elegans* from L1 stage to Adult (Day 2) stage. **b** Metabolic heat output measurements of N2 wild-type worms (*n* = 61 independent trials) at different stages from L1 to Adult (Day 4). The red and blue triangles represent the AMR and SMR of individual worms, respectively, and are connected by a black dashed line to identify measurements from individual worms. The inset shows the mass-specific AMR (red) and SMR (blue) averaged over 61 N2 wild-type worms while the error bars represent the standard deviation (±SD) about the mean. **c** SMR vs. body size on a log-log plot to extract an allotropic scaling constant of 0.76 from (*n* = 52 independent trials) wild-type N2 samples from L3 to Adult (Day 3). The purple solid line represents the best-fit with R-squared error of 0.75. The purple shaded region indicates the 95% confidence interval.

stages (L3–Day 3), which concurs with the reported inter-specific value of $0.72 \pm 0.09$[31]. Thus, our work shows that the above scaling law is applicable to adult *C. elegans* intra-specifically and predicts the mass-specific metabolic rate across different stages.

**A long-lived mutant, daf-2, has lower metabolic heat output.** To further demonstrate our system's ability to detect differences in metabolic activities, we have performed age-dependent metabolic heat measurements on a long-lived mutant, *daf-2 (e1370)*. It is well established that the mutations that reduce the activity of a gene called *daf-2* (insulin/insulin-like growth factor-1 receptor) double the worm's lifespan[16]. This gene, which controls the expression of multiple longevity genes, is known to affect several biological processes including development, metabolism, and resistance to stress. Assaying physiological factors like metabolic rates of such mutants in addition to the chronological lifespan measures could provide further insights into aging mechanisms[9]. In our studies, both the strains (N2 wild-type and *daf-2* mutant) are cultured and maintained at identical conditions (room temperature 22 °C) and care was taken to perform the measurements under the same environmental conditions (inner shield temperature 25 °C, flow rate 100 nl min⁻¹) for a consistent comparison. We performed metabolic rate measurements on adult wild-type and *daf-2* mutant worms as they age from Day 1–4 and observed significant difference in metabolic heat outputs (wild-type: $92.2 \pm 17$ nW, *daf-2*: $53.9 \pm 8.7$ nW) between both genotypes (Fig. 5a). In the case of wild-

type, an increase in AMR from $78.3 \pm 11.6$ nW to $108.1 \pm 9.1$ nW is observed as the worm progresses from Day 1–4 (Fig. 5b). The metabolic rates of *daf-2* reveal a significant metabolic shift to lower rates with Day 1 worms showing an AMR of $61.8 \pm 5.8$ nW, which further drops to $45 \pm 5.8$ nW on Day 4. We note that the variations ($\pm 12$ nW maximum) observed in the metabolic heat outputs of worms of the same age may result from the limited duration from which the AMR is determined, differences in the time of day when the measurements were taken or individual physiological variability, which further highlights our instrument's capability to discern such differences. Based on the size measurements from CCD images, we also report age-dependent mass-specific AMR (Fig. 5c) of the same sample set described above where it is known that metabolic rate normalized with protein content is negatively correlated with age of the adult worms[27]. We observe that the mass-specific AMRs of Day 1 wild-type and *daf-2* worms are significantly different with average values of $69.3 \pm 9.6$ µW mg⁻¹ and $36.1 \pm 5.7$ µW mg⁻¹, respectively, which further reduced to $42.9 \pm 3.8$ µW mg⁻¹ and $27.8 \pm 7.3$ µW mg⁻¹ on Day 4. The rate of decrease in mass-specific AMR of *daf-2* ($\sim -2.7$ µW mg⁻¹ day⁻¹) is significantly small compared with that of wild-type ($\sim -9$ µW mg⁻¹ day⁻¹), suggesting a potential mechanism for increased longevity.

**Discussion**
We have performed direct, high-resolution calorimetric heat measurements and related them to the physiological state of individual

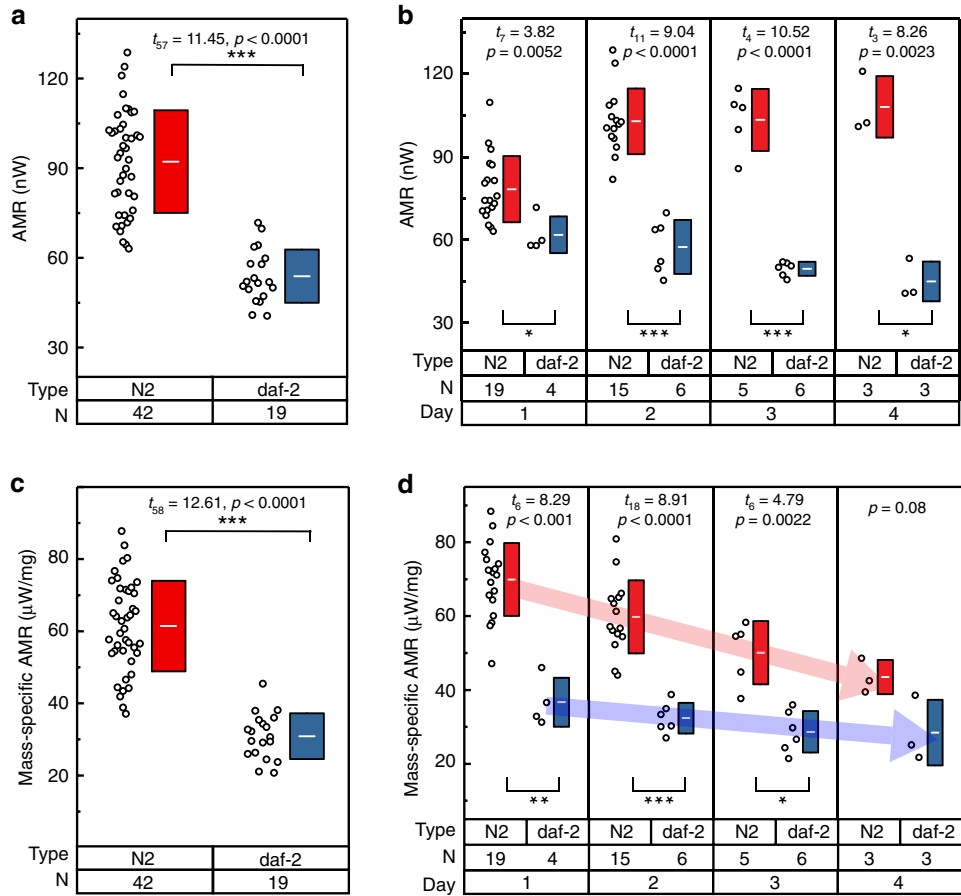

**Fig. 5 Age-dependent metabolic heat output measurements on N2 wild-type and daf-2 mutant. a** AMR from 42 worms of wild-type and 19 worms of *daf-2* from Day 1–4. The white line in the center of the box represents the mean value of the corresponding set, whereas the height of the box represents the standard deviation (±SD). N indicates the corresponding number of samples. **b** AMR of several worms (black circles) of wild-type (red) and *daf-2* (blue) from Adult Day 1–4. **c** Mass-specific AMR of the sample set in Fig. 5a. **d** Mass-specific AMR is plotted for the same sample set in Fig. 5b. Statistical analysis was performed using two-tailed t test. Statistical significance was determined for $p < 0.05$; *$p < 0.01$, **$p < 0.001$, ***$p < 0.0001$.

*C. elegans* via simultaneous optical measurements. Cellular processes (e.g., energy conversion, gene expression, signal transduction, cell division) are not 100% efficient and are always accompanied by heat dissipation. Therefore, the proposed continuous opto-calorimetric measurements from individual organisms can enable direct heat output quantification of the metabolic contributions of various intracellular processes. This sub-nW resolution (~270 pW with a temporal resolution of 100 s) direct calorimeter was accomplished by a combination of low thermal conductance capillary tubes ($G_{th}$ ~ 27 $\mu$W K$^{-1}$) and high-resolution ($\Delta T$ ~ 10 $\mu$K) thermometry. This resolution presents more than two orders of magnitude improvement over the best calorimeters previously employed[19] in *C. elegans* studies and one order of magnitude resolution improvement over the most-sensitive calorimeter[20] employed to date for biological studies. We demonstrate the usefulness of this instrument for studies on *C. elegans* by conducting a series of measurements on larval and adult stages. In fact, this work represents the first metabolic rate measurement on a single *C. elegans* from L1 to the adult stage. In addition, our calorimeter captures variations of metabolic heat generation corresponding to the worm's locomotive activity, which is shown to be ~5–25% above the SMR. Further, the worm's metabolic responses to external stimuli, like temperature and oxygen deprivation, may also be explored in our system. This method can be combined with other metabolomics approaches where mass-spectrometry can be used to profile metabolites in cells. Calorimetric approaches overcome some of the limitations posed by metabolomics approaches: for example, destructive sample preparation prevents continuous, time-resolved measurements, and relating detected biomarkers to biological mechanisms is challenging. Finally, we note that our calorimeter, built from commercially available parts, without involving any complicated microfabrication processes, can in principle be employed to study metabolism of several other model organisms (e.g., Chlamydomonas, brown fat cells) whose metabolic output is in the nanowatt range.

## Methods

**Fabrication of the calorimeter.** Each capillary is a 20 mm long, hollow, borosilicate capillary tube (250 × 250 $\mu$m, wall thickness 50 $\mu$m) coated with 100 nm-thick gold on all sides except a small portion (~2 mm) in the center to provide optical access for imaging. A 125 $\mu$m-diameter borosilicate rod is placed close to the center for localizing the *C. elegans* movement to the center of the capillary tube (Fig. 1d). The borosilicate capillary tube has the advantage of transparency for optical imaging and low thermal conductivity, whereas the square cross-section with the flat surface aids in imaging by avoiding the aberrations owing to the curvature.

A schematic of the fabrication process for the calorimetric system is shown in Supplementary Fig. 2. The tubes (VitroTubes$^{TM}$) are first cleaned in an oxygen plasma cleaner to remove organic contaminants and improve surface adhesion for further processing. They are then masked with Kapton tape before being sputtered with Ti/Au (10/100 nm) thin films in the next two steps. The first mask (~400 $\mu$m across) in the center provides the isolated electrical leads, whereas the second mask (~2 mm) on the opposite surface generates a window for optical access. The Au layer thickness is chosen to reduce the radiative coupling of the tube to the outer shield while minimizing the thermal conductance along the tube and maintaining a high quality film for uninterrupted electrical current flow. A 10 kΩ thermistor (Murata Electronics, −4% TCR, 0603 metric) is then soldered between the two leads. The tubes are flipped and mounted on the IS, where the electrical isolation between the IS and tubes is achieved by incorporating a 100 $\mu$m-thin glass slide while silver epoxy is used to improve thermal contact. The 125 $\mu$m diameter glass fiber is then inserted into the capillary tube and fixed at the exposed end with epoxy. Finally, a PEEK tube (360 $\mu$m/150 $\mu$m OD/ID) is connected to the capillary tube through a PEEK tubing sleeve (800 $\mu$m/400 $\mu$m OD/ID) and all the gaps are sealed with a vacuum epoxy. This tubing system is then anchored to the MS and passed through microfluidic connectors (MicroTight Adapter PEEK 1/16" ID × 360 $\mu$m w/Fittings) on the OS that act as vacuum feedthroughs to the outside.

**High stability temperature system.** To improve the temperature resolution, we built a three-shield system (Fig. 1a, b) with large thermal time constants to decouple high frequency temperature fluctuations from the ambient and controlled individual shield's temperature in a PID loop to reduce low frequency temperature fluctuations. The OS is a nested three-shield structure forming a vacuum enclosure

(20 × 20 × 12 cm$^3$, wall thickness of 1.2 cm) with electrical, optical, and fluidic feedthroughs holding vacuum down to at least 10 $\mu$Torr. Thermistors (US Sensor Corp. USP12838) are bonded to each shield in a drilled hole at a representative location using epoxy (3M Scotch-Weld Epoxy Adhesive 2216 B/A) and provide temperature feedback for PID control. Polyimide flexible heaters (Omega KH series) are attached in series to each shield on different surfaces providing uniform heat to the system. The MS (7 × 7 × 3.5 cm$^3$) with a central pocket (4 × 4 × 2 cm$^3$), is held on the bottom part of OS supported by four spherical borosilicate balls. The IS (3 × 3 × 1.2 cm$^3$) with a central pocket (2 × 0.8 × 0.6 cm$^3$), is then suspended in the MS pocket by three polymer supports with a spherical end. These spherical contacts provide weak thermal links (OS/MS–50 mW K$^{-1}$ & MS/IS–5 mW K$^{-1}$) while maintaining robust mechanical stability. From the thermal mass of copper shields and the calculated thermal conductance, the thermal time constants of the OS, MS, and IS are estimated to be 3000 s, 12,000 s, and 1600 s, respectively, which match with the time constants obtained experimentally. The PID control scheme, as described in the following section, enables our setup to maintain OS, MS, and IS temperatures to within ±1 mK, ±15 $\mu$K, and ±15 $\mu$K (Supplementary Fig. 6), respectively.

**Thermometry and temperature control of shields.** In order to measure temperature changes with $\mu$K resolution, we used resistance-based thermometry in an AC-driven Wheatstone bridge configuration (Supplementary Fig. 3a). In brief, the right half of the bridge consists of a sensing resistor on the lower branch with an associated resistance on the top, whereas the left side is described as the matching side. Fixed resistors with ultra-low temperature coefficient of resistance (Vishay Z201 Series Z-Foil Resistors, ±0.2 ppm K$^{-1}$) are used with the resistance values chosen to improve the stability and resolution based on previous studies[25,32]. To balance the bridge, a fixed matching resistance equal to that of the sensing thermistor's resistance at its average temperature of operation is used, whereas a potentiometer (Vishay Spectrol 534 series, ±20 ppm K$^{-1}$) is used on the top branch of the matching side for fine tuning. The bridge is excited by a sinusoidal voltage ($V_{AC}$ = 1 V peak-to-peak) using a waveform generator (Agilent 33210) at frequencies in the range of 10–100 Hz (frequencies were selected to avoid crosstalk). We note that this sinusoidal voltage dissipates ~0.5 $\mu$W in the calorimeter, causing the sensing capillary temperature to rise by 20 mK above the baseline temperature without sinusoidal excitation. The matching and the sensing signals are then fed into an instrumentation amplifier (Analog Devices AD524) where the common-mode signal is subtracted, and differential mode AC signals are amplified (Gain = 100 and Gain drift is at most 25 ppm K$^{-1}$). The amplitude of this amplified signal is then measured in a lock-in scheme (SRS 830) at a bandwidth of 0.1 Hz and recorded using a data acquisition card (PCI-6014) in the LabVIEW environment. The temperature resolution of the resulting circuit is quantified to be ±5 $\mu$K (Supplementary Fig. 4). As shown in Fig. 2c, $T_{sen}$ and $T_{mat}$ are the temperatures of the sensing and the matching capillary tube and are measured by employing the corresponding thermistors in two independent circuits based on the thermometry described above.

To provide current to the polyimide heaters introduced above, we developed a voltage-controlled current source (Supplementary Fig. 3b) where the emitter voltage of the transistor is controlled through an op-amp with the polyimide heater on the collector side. This op-amp is configured to sum a manual voltage signal ($V_{man}$) providing heat to reach the desired temperature and a controlled voltage signal ($V_{PID}$) to stabilize the temperature at its set-point in a PID loop (Supplementary Fig. 3c). The temperature measured from the Wheatstone bridge circuit acts as feedback to the PID controller (implemented in LabView 2018) whose output controls $V_{PID}$. Preliminary PID parameters were obtained from the step response of each thermal system using SIMC criterion[33] and were further manually tuned to minimize the temperature fluctuations.

**Optical imaging.** Our calorimeter incorporates good optical access to the central region of the sensing capillary tube, which aids in trapping and monitoring the activity of the *C. elegans*. Though systems employing multiple shields to achieve thermal stability are well established, integrating an optical imaging capability is challenging as this naturally couples environmental fluctuations into the system through radiative coupling and the illumination source. A combination of IR reflective (Hot Mirror-Edmund Optics) and IR absorptive (Heat absorbing KG5 Schott filter) windows, sealed to the bottom part of the OS (Fig. 1b) reduce the coupling to the outer environment by reflecting IR radiation (~90%) to the environment and absorbing the remaining transmitted radiation. The temperature-controlled calorimeter is mounted on an inverted microscope (ZEISS Axiovert 200) and imaging is performed with epi-illumination through a 10x objective (CFI Plan Fluor DL 10XF). A DC power supply (Agilent 6033 A) provides power (3 V, 0.75 A) to the Halogen light source while a CCD camera (RET-4000R-F-M-12-C) controlled through Micro-Manager 1.4.22 acquires images at one frame per second. The minimum illumination that provides a visible image at an exposure time of 0.5 s is chosen to minimize thermal instabilities owing to the light source fluctuations. This illumination results in a temperature rise of ~1 mK (corresponding to ~27 nW) on both sensing and matching capillary tubes. We note that all thermal signals reported here were measured under this illumination setting.

**Estimating the size and activity of C. elegans.** The mass/volume of an individual *C. elegans* needed for obtaining the mass-specific properties was estimated from the images captured during the measurement. Considering the axi-symmetric structure of the worm, the body is carefully divided into small sections of cylinders[34] along the length of worm (see Supplementary Fig. 5a). The relevant dimensions (pixel lengths) for each section (diameter $D_n$ and width $H_n$) were measured in ImageJ with the appropriate calibration of the microscope's magnification. The volume of the worm is the sum of the volumes of all cylindrical sections. The mass of each worm is then derived from the total estimated volume with the density considered to be ~1.08 g/cc[35].

$$\text{Volume} = \sum_{n=1}^{N} \frac{\pi D_n^2 H_n}{4}$$

An activity factor is defined to quantitatively represent the worm's activity, which is derived by performing the following image analysis in ImageJ on a stack of images collected during the measurement at a frame rate of 1 s. Initial brightness and contrast adjustments are performed on two image stacks, where the second image stack is a duplicate of the first with a single frame offset. The two stacks are then subtracted to create a new stack, which now represents the worm activity, i.e., subtracting a frame at $(n + 1)^{th}$ sec (Supplementary Fig. 5b) from that at the $n^{th}$ sec eliminates the motionless background and captures only the movement of the worm. To quantify this movement, the mean pixel intensity of each frame was then calculated and converted to a dimensionless activity factor ranging from 0 to 1, where 0 and 1 indicate the lowest and the highest activity levels of a given worm. We note that absolute quantities can be used for activity comparison between wild-type and *daf-2* worms, instead of normalized quantities.

**Thermal conductance measurement and power resolution.** To experimentally quantify the thermal conductance of our capillary system, DC power is input at the center of the capillary tube by superimposing a DC voltage offset ($V_{DC}$) on top of the sensing AC voltage ($V_{AC}$), which causes a DC temperature rise ($T_{DC}$) of the sensing capillary tube corresponding to the power expenditure of $P_{DC} = \frac{V_{DC}^2}{R_{th}}$, where $R_{th}$ is the nominal electrical resistance of the thermistor at room temperature (~10 kΩ). The resultant temperature rise is measured by the same thermistor using the thermometry scheme discussed above with a bandwidth of 5 mHz. The slope of $P_{DC}$ with respect to $T_{DC}$ corresponds to the thermal conductance $G_{th}$ of the capillary system. As shown in Fig. 2a, we dissipated power in the calorimeter from 10 nW to 100 nW in steps of 10 nW and measured corresponding temperature rises from 0.37 mK to 3.7 mK. From the slope of this plot, we estimate the thermal conductance of the capillary tube to be 27 µW K$^{-1}$. Further, to test the response of our calorimeter to sub-nW inputs, we varied the offset voltage ($V_{DC}$) from 1 mV to 3 mV in steps of 0.2 mV. This corresponds to a power dissipation of ~100 pW to ~1 nW in the calorimeter. In the inset of Fig. 2a, we plot this temperature rise with respect to the power dissipation. The data points shown in the plot correspond to mean values of four independent measurements performed for each power input, whereas the error bars represent standard deviation in the temperature rise. From the observed standard deviation of 5 µK, we estimate the noise floor to be 10 µK (the maximum standard deviation observed at different heat inputs). Thus, the minimum precisely measurable power change using our calorimeter is defined as the noise floor and is estimated to be 270 pW (corresponding to temperature drift of ~10 µK). We note that the theoretical electronic noise in our system considering Johnson noise, shot noise, voltage source noise, and the noise added at the amplifier inputs corresponds to a NET of <1 µK in a bandwidth of 5 mHz. But the inherent long-term drift in our system, unintended radiative coupling to the outer shield and added noise in the data acquisition result in the increased NET of 10 µK (270 pW). Thus, a resolution of <100 pW is potentially possible by carefully addressing long-term drift. Further, to establish the sub-nW resolution of our system, a DC offset voltage ($V_{DC}$) of ~1.6 mV, corresponding to power dissipation of 270 pW (thermal pulse in Fig. 2b) is applied and the resulting steps in temperature rise of 10 µK can be clearly observed in Fig. 2b. Similarly, a square pulse with an amplitude of ~16 mV, corresponding to a power dissipation of 27 nW, is applied to estimate the time constant of our system to be 100 s (Fig. 2d).

**Sample preparation and control.** The nematodes were maintained at 20 °C on NGM agar plates with OP50 *Escherichia coli* as food source, according to standard protocols[36]. The following strains were used in our studies: Bristol N2 (used as wild-type), *daf-2(e1370)*. For heat output measurements, the worms were transferred to the sensing capillary tube at 25 °C, which was filled with S-basal buffer.

**Data analysis and statistics.** Data analysis was performed in Excel 2016 and Origin 2019b. Statistical analysis reported in Fig. 5 was performed using a two-tailed *t* test in Origin 2019b and the resulting statistics of t-statistic, degrees of freedom (subscript of t-statistic), *n* (number of samples), and *p* values > 0.0001 are reported.

**Reporting summary.** Further information on research design is available in the Nature Research Reporting Summary linked to this article.

## Data availability

The data that support the findings of this study are available from the corresponding authors on reasonable request.

## Code availability

Custom code used in this work is available upon reasonable request.

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

## Acknowledgements

This work was supported by a University of Michigan M-Cubed grant to S.Y., P.R., and E.M., and a National Institutes of Health NIGMS grant (R35GM133737) to S.Y. We thank professor Scott Leiser and professor Nicholas Chronis for sharing worm strains and lab resources for culturing *C. elegans*.

## Author contributions

P.R. and E.M. conceived the project. S.H. and R.M. designed the experiments under the supervision of S.Y., P.R., and E.M. The calorimetric setup was designed and built by S.H. and R.M. under the supervision of P.R. and E.M. The worm strains used in the experiments were maintained and prepared by S.Y. The paper was written by all authors.

## Competing interests

The authors declare no competing interests.
