## [Peer Review File · Nature Communications]

Reviewers' Comments:

Reviewer #2:

Remarks to the Author:

The authors present the design and construction of a sensitive calorimeter which allows accurate quantification of heat release from small organisms suspended in a liquid environment. The calorimeter is compatible with simultaneous microscopic visualization of the test subject. The presented work is scientifically sound and accurate. I only have minor remarks.

Abstract

- "...we show that the metabolic output is significantly lower in *C. elegans* mutants with greater life expectancy." This is a generalization; the authors test only one long-lived mutant [*daf-2(e1370)*] in this study. Other mutants may show different results.

Introduction

- Page 3, 1st line: "...approach for accurate characterization of metabolism...". The metabolism is not characterized, it is quantified. Calorimetry alone does not allow for discrimination of metabolic pathway activity.

Results

- First sentence of the section 'Design of the calorimeter': provide references that support the figures (500 and 10-fold improvement).
- Figure 2a: how is the noise floor of 10 μ K determined?
- Section 'Heat output protocol and analysis': mention the temperature at which the metabolic measurements are carried out. Is that 25°C, as mentioned elsewhere in the manuscript? It must be noted that 25°C causes borderline heat stress in *C. elegans*. The standard temperature for *C. elegans* culture is 20°C. Is the microcalorimeter sensitive enough to measure the reduced worm metabolism at 20°C as well?
- Page 10 and elsewhere in the manuscript: BMR (basal metabolic rate) is used only for endotherms (while fasting and resting in the thermoneutral zone). *C. elegans* is an ectothermic organism, hence the term standard metabolic rate (SMR) should be used (with mentioning of the temperature at which the measurement is performed).
- Figure 3a: can the sudden temperature shift that occurs during loading and unloading (blue areas of the curve) be explained?
- It is calculated that AMR is about 7% higher than BMR. This number seems to deviate significantly to the difference between AMR and SMR estimated by Laranjeiro et al. (PMID: 28395669). Can the authors explain this discrepancy? Do the authors suggest that the aerobic scope of *C. elegans* is likely extremely small? This should be discussed in the manuscript.
- Page 14, line 9: avoid the term 'developmental stages' when referring to fully grown adult *C. elegans*. This is confusing as 'development' is commonly used for stages L1 to L4 (also in legend of Fig. 5).
- Page 14, line 14: "We note that the variations...observed in the metabolic heat outputs of worms of the same age may result from the limited duration from which AMR was determined, differences in the time of day when the measurements were taken, or individual physiological variability, which further highlights our instruments capability to discern such differences." This suggests that the authors know that the source of variation is of biological nature, not of technical nature (or random noise). How can they distinguish between the two?
- Page 15 line 1: italicize *daf-2*.
- From a biological perspective, Figure 5 is the most interesting one in this manuscript. The authors show absolute and biomass-corrected AMRs of wild-type and *daf-2* mutants. Why did they opt to show AMR rather than SMR? It is well known that *daf-2* mutants are generally less active compared to wild-type. It would be interesting to see whether the lower AMR of young *daf-2* mutants can be completely attributed to the slow phenotype or not. Are activity measurements available for this data set as well? This would allow to directly correlate the age-dependent decline in activity to metabolic rate.

Discussion

- I recommend the authors to be more explicit in the discussion section on the potential use of this instrument with other biological systems. Could, for instance, *Drosophila* larvae be tested in a non-liquid environment (in some kind of flow-and-drain setup to load the animals)?

Reviewer #3:

Remarks to the Author:

This paper is interesting to focus the thermal measurement of large-scale objects with small-heat amount generations.

*The focusing point is not clear. To develop the calorimeter of sub-nanowatt resolution or to clarify the metabolic characteristics of *C. elegans*? If former case, I think the calorimeter seems to be combination of already-established technique. If not, you should explain what is novel techniques in your developed calorimeter.

If later case, you should explain in detail, adding to "*C. elegans* metabolism is being actively investigated due to its promise for providing insights into human disease and ageing", not only show the reference papers.

*Would you indicate the size of the developed calorimeter and *C. elegans*? i.g. in the sentence or in the image as a scale bar.

*How did you measure the temperature stability of the shields? How many measurement points was on the shields?

*What is the meaning of the weak heat links between the shields?

* How was the thermal pulse input in Fig.2?

Response Letter

We thank the reviewers for taking the time to review our manuscript and providing insightful comments. To facilitate reading of this response letter, we have used blue for the reviewer comments while our response is written in black.

Reviewers' comments:

Reviewer #2 (Remarks to the Author):

The authors present the design and construction of a sensitive calorimeter which allows accurate quantification of heat release from small organisms suspended in a liquid environment. The calorimeter is compatible with simultaneous microscopic visualization of the test subject. The presented work is scientifically sound and accurate. I only have minor remarks.

Abstract

- "...we show that the metabolic output is significantly lower in *C. elegans* mutants with greater life expectancy." This is a generalization; the authors test only one long-lived mutant [*daf-2(e1370)*] in this study. Other mutants may show different results.

We agree with the reviewer that our original claim was not specific enough, as we have tested only one long-lived mutant, *daf-2*. Accordingly, we have modified the manuscript to read: "Further, we show that the metabolic output is significantly lower in long-lived *C. elegans daf-2* mutants."

Introduction

- Page 3, 1st line: "...approach for accurate characterization of metabolism...". The metabolism is not characterized, it is quantified. Calorimetry alone does not allow for discrimination of metabolic pathway activity.

We have changed this section to "...a non-invasive approach for accurate quantification of metabolic activity,"

Results

- First sentence of the section 'Design of the calorimeter': provide references that support the figures (500 and 10-fold improvement).

References that support our claim have been included in the revised manuscript. Specifically, the first sentence now reads as: "Our novel calorimeter provides an unprecedented heat resolution of ~270 pW—a 500-fold¹⁹ improvement over calorimeters employed for past *C. elegans* studies and a 10-fold^{20,21} improvement compared to the state-of-the-art bio-calorimeters."

Where the cited references are as follows:

(19) Krenger, R., Lehnert, T. & Gijs, M. A. M. Dynamic microfluidic nanocalorimetry system for measuring *Caenorhabditis elegans* metabolic heat. *Lab Chip* **18**, 1641-1651 (2018).

(20) Inomata, N., Toda, M. & Ono, T. Highly sensitive thermometer using a vacuum-packed Si resonator in a microfluidic chip for the thermal measurement of single cells. *Lab Chip* **16**, 3597-3603 (2016).

(21) Lee, W., Fon, W., Axelrod, B. W. & Roukes, M. L. High-sensitivity microfluidic calorimeters for biological and chemical applications. *Proceedings of the National Academy of Sciences USA* **106**, 15225-15230 (2009).

- **Figure 2a: how is the noise floor of 10 μ K determined?**

In Fig. 2a (inset) we present data from an experiment where the power input to the calorimeter was varied in sub-nW increments while simultaneously recording the temperature change from the embedded thermistor. The data points shown in the plot correspond to mean values of four independent measurements performed for each power input whereas the error bars correspond to the standard deviation across the four measurements. It can be seen that in some of these measurements the error bars are as large as $\pm 5 \mu$ K (which we report as a noise floor of 10 μ K). These errors arise largely from slow drift in the calorimeter temperature. We note that a careful analysis of electronic noise sources (Johnson noise, shot noise and amplifier noise) suggests that their contribution to the noise floor is smaller ($<1 \mu$ K), suggesting that the measurements reported in the manuscript are limited by long-term temperature drift of our temperature shields.

In order to clarify this issue to future readers we have modified both the figure caption of Fig. 2a and our discussion in the Methods section titled “Thermal conductance measurement and power resolution”.

- **Section ‘Heat output protocol and analysis’: mention the temperature at which the metabolic measurements are carried out. Is that 25°C, as mentioned elsewhere in the manuscript? It must be noted that 25°C causes borderline heat stress in *C. elegans*. The standard temperature for *C. elegans* culture is 20°C. Is the microcalorimeter sensitive enough to measure the reduced worm metabolism at 20°C as well?**

We thank the reviewer for this important question. Our calorimeter can certainly be modified to measure metabolism at 20°C and would be sufficiently sensitive to perform such measurements.

All measurements were performed at 25°C. For clarity, we have now included this in the beginning of the section, to read: ‘we performed metabolic measurements on individual *C. elegans* at 25°C’. We decided to perform measurements at 25°C as some previous studies [Houthoofd, K. et al, *Aging Cell* **4**, 87-95 (2005); Houthoofd, K. et al, *Neurobiology of Aging* **26**, 689-696 (2005)] have also conducted metabolic measurements under similar conditions. Another reason for this choice was that for our calorimeter temperature stabilization was performed using heaters, which enables stabilization only at temperatures above room temperature (22°C, in our case).

If measurements at lower temperature are desired, the heaters can be replaced with Peltier coolers to achieve cooling. We anticipate no significant challenges in resolving metabolic

activities at these slightly lowered temperatures as the calorimetric resolution of our instrument will still be ~ 270 pW, while the metabolic output of the individual worms is expected to be at the level of a few, even at 20°C [see V. Voorhies and S. Ward, *PNAS*, 96 (20) 11399-11403 (1999)], which is well above our resolution limit.

- Page 10 and elsewhere in the manuscript: BMR (basal metabolic rate) is used only for endotherms (while fasting and resting in the thermoneutral zone). *C. elegans* is an ectothermic organism, hence the term standard metabolic rate (SMR) should be used (with mentioning of the temperature at which the measurement is performed).

We thank the reviewer for this comment. To be consistent with the nomenclature for ectotherms, we have changed this term to SMR indicating the temperature at which measurement is performed (25°C), instead of BMR.

- Figure 3a: can the sudden temperature shift that occurs during loading and unloading (blue areas of the curve) be explained?

Yes, the transient temperature shift arises from the change in the effective thermal conductance of the capillary tube due to an increase in flow rate of the fluid to 2-10 $\mu\text{l}/\text{min}$ from the nominal flow rate of 100 nl/min . To elaborate, the transient temperature shifts during loading/unloading can be quantified by the simple thermal model of our measurement as shown in Figure R1 below. The chamber is thermally connected to inner shield (IS) at a nominal temperature of $\sim 24.5^\circ\text{C}$ (T_2) and the outer shield (OS through radiative coupling) at a temperature of $\sim 24.38^\circ\text{C}$ (T_1). The resulting temperature change is described by the equation below:

$$T = \frac{Q + G_{\text{rad}}T_1 + G_{\text{cond}}T_2}{G_{\text{rad}} + G_{\text{cond}}},$$

where Q is the heat produced by an individual *C. elegans*. In the initial reference case (gray region in Fig. 3a) at a flow rate of 100 nl/min , $G_{\text{rad}} = 6 \mu\text{W}/\text{K}$ and $G_{\text{cond}} = 21 \mu\text{W}/\text{K}$. In the absence of *C. elegans*, we have a power dissipation of $Q = 0.5 \mu\text{W}$ in the thermistor as a result of the AC excitation employed in our thermometry scheme (discussed in detail in Methods section titled Thermometry & Temperature control of shields). This leads to the initial chamber temperature (T) of 24.492°C , slightly below that of the inner shield. During loading and unloading, the high flow rates increase G_{cond} drastically. For example, we expect the G_{cond} to increase to 101 $\mu\text{W}/\text{K}$ at a flow rate of 4 $\mu\text{l}/\text{min}$. For this conductance value, the resulting temperature is $\sim 24.497^\circ\text{C}$, a 5 mK increase above the initial reference. Thus, depending on the flow rates used (2-10 $\mu\text{l}/\text{min}$), a temperature rise on the order of a few mK can be expected, as evident in the blue regions of Fig. 3a (loading/unloading). The increase in temperature at the end of the first blue region (loading stage) indicates the entry of *C. elegans* into the capillary tube, at which point the flow rate is reduced to 100 nl/min . The transient following this change is indicative of the thermal time constant of our calorimeter.

Figure R1. Thermal model of the calorimeter: The temperatures of chamber, outer shield (OS) and inner shield (IS) are represented by T , T_1 and T_2 respectively. G_{cond} and G_{rad} represent the thermal conductances of the coupling to IS and the OS. Q is the heat produced by a specimen in the chamber.

- It is calculated that AMR is about 7% higher than BMR. This number seems to deviate significantly to the difference between AMR and SMR estimated by Laranjeiro *et al.* (PMID: 28395669). Can the authors explain this discrepancy? Do the authors suggest that the aerobic scope of *C. elegans* is likely extremely small? This should be discussed in the manuscript.

We thank the reviewer for this important comment. The SMR measured by Laranjeiro *et al.*, (*BMC Biology*, **15**, 30 (2017)) is about 50% of AMR, as opposed to the 7% change observed in our study. First, we would like to point that a direct comparison of our results with those of Laranjeiro *et al.* is challenging due to important differences in the experimental conditions. Laranjeiro *et al.* acquired standard metabolic rate (SMR) of animals, immobilized either by levamisole-induced paralysis of wild-type N2 animals or by the genetic paralysis of *unc-54* mutants deficient in a muscle myosin heavy chain. Levamisole causes worms to stay shrunk for several hours at steady-state contraction. This is evidently different from our measurement conditions, which were conducted on freely moving *C. elegans*. We did not restrain worm's activity to monitor metabolic rate; in fact, we specifically selected only those periods during metabolic rate measurements when worms were resting (no locomotion for a short time) to calculate SMR. To our knowledge, this is the first measurement of SMR on unconstrained worms, making it challenging to compare with metabolic heat rates or oxygen consumption rates from previous reports. We note that the question raised by the reviewer is very interesting and needs more controlled experiments like metabolic rate measurements on paralyzed worms, to definitively explain the discrepancy, which would have been addressed in future studies.

- Page 14, line 9: avoid the term 'developmental stages' when referring to fully grown adult *C. elegans*. This is confusing as 'development' is commonly used for stages L1 to L4 (also in legend of Fig. 5).

We thank the reviewer for this suggestion. We have now specified the age of adult *C. elegans* (Day 1 – Day 4) where appropriate.

- Page 14, line 14: “We note that the variations...observed in the metabolic heat outputs of worms of the same age may result from the limited duration from which AMR was determined, differences in the time of day when the measurements were taken, or individual physiological variability, which further highlights our instruments capability to discern such differences.” This suggests that the authors know that the source of variation is of biological nature, not of technical nature (or random noise). How can they distinguish between the two?

We believe that such variations are of biological nature, due to the following reasons:

- 1) The random noise in our system, which is ~270 pW (see Fig. 2a), is much smaller than the variations observed in the metabolic heat outputs (on the order of tens of nW).
- 2) The initial reference value (gray region in Fig. 3a) stabilizes back to the same level after several loading/unloading trials. A nominal drift of 10 μ K, corresponding to 270 pW, was observed over a day, which is still far below the variations observed. No spurious jumps in the signals could be observed in such measurements over 30 hours (Figure 2C). This confirms that the presence of the specimen (here *C. elegans*) is what causes these variations.

- Page 15 line 1: italicize *daf-2*.

We thank the reviewer for pointing out this error. This has been corrected in the manuscript.

- From a biological perspective, Figure 5 is the most interesting one in this manuscript. The authors show absolute and biomass-corrected AMRs of wild-type and *daf-2* mutants. Why did they opt to show AMR rather than SMR? It is well known that *daf-2* mutants are generally less active compared to wild-type. It would be interesting to see whether the lower AMR of young *daf-2* mutants can be completely attributed to the slow phenotype or not. Are activity measurements available for this data set as well? This would allow to directly correlate the age-dependent decline in activity to metabolic rate.

We thank the reviewer for these interesting questions. It has been our observation that both, wildtype and *daf-2* mutant worms, were generally active during the entire duration of the measurement, making it difficult to extract SMR from such measurements. Since the aim of this study is to establish our tool’s capabilities, we presented only the AMR of *daf-2* mutant worms to allow comparisons with previous reports. We will explore the above described questions in detail in our future studies.

Discussion

- I recommend the authors to be more explicit in the discussion section on the potential use of this instrument with other biological systems. Could, for instance, *Drosophila* larvae be tested in a non-liquid environment (in some kind of flow-and-drain setup to load the animals)?

We have now significantly enhanced the discussion section. As mentioned in the discussion section, metabolic heat outputs of several other biological systems such as brown fat cells, Tetrahymena, Chlamydomonas can be readily measured in our system. We note that measurements on Drosophila larvae would not be possible with our current instrument, mainly due to the size of these larvae (few millimeters) being much larger than the capillary size (150 μm) employed in this system. We note that the system developed by us in the past (Fiorino, A. *et al.*, Parallelized, real-time, metabolic-rate measurements from individual Drosophila, *Scientific Reports* 8, 14452 (2018)) in a non-liquid environment, is more suitable for measurements on Drosophila larvae.

Reviewer #3 (Remarks to the Author):

This paper is interesting to focus the thermal measurement of large-scale objects with small-heat amount generations.

*The focusing point is not clear. To develop the calorimeter of sub-nanowatt resolution or to clarify the metabolic characteristics of *C. elegans*? If former case, I think the calorimeter seems to be combination of already-established technique. If not, you should explain what is novel techniques in your developed calorimeter.

If later case, you should explain in detail, adding to “*C. elegans* metabolism is being actively investigated due to its promise for providing insights into human disease and ageing“, not only show the reference papers. [ED: Reviewer 1 has expertise in *C. elegans* metabolism and was satisfied with the level of detail you provided here].

Our focus is to develop a robust, highly sensitive and bio-compatible calorimetric platform with the capability to measure heat outputs of several biological models such as *C. elegans*. To our knowledge, this is the first demonstration of metabolic heat measurements with such a high-resolution, which is achieved through a novel combination of established techniques and extremely careful experimental implementation. Although we present heat output measurements on *C. elegans* in this study, we note that our calorimeter can be generally applied to other systems such as Brown fat cells, Tetrahymena etc.

With regards to the detailed explanation adding to “*C. elegans* metabolism is being actively investigated due to its promise for providing insights into human disease and ageing“, we believe more details explaining and corroborating this phrase have been provided in two locations:

Second paragraph of ‘Introduction’ section:

“For example, it is well known that genetic mutations, such as *age-1* and *daf-2* mutants, increase the lifespan of the worms mediated via various transcription factors with roles in insulin signaling, autophagy, and cellular energy metabolism¹⁶⁻¹⁸. Thus, metabolic heat output measurements on single worms through direct calorimetry can provide fundamental insights into metabolic pathway regulation in context to the above biological mechanisms”

First paragraph of ‘Comparison of metabolic heat output measurements on N2 and *daf-2*, a long-lived mutant’ section:

“This gene, which controls the expression of multiple longevity genes, is known to affect several biological processes including development, metabolism and resistance to stress. Assaying physiological factors like metabolic rates of such mutants in addition to the chronological lifespan measures could provide further insights into aging mechanisms⁹”

*Would you indicate the size of the developed calorimeter and *C. elegans*? i.g. in the sentence or in the image as a scale bar.

We thank the reviewer for the comment. We now added the scale bar in Figure 1.

***How did you measure the temperature stability of the shields? How many measurement points was on the shields?**

Several thermistors were mounted on each shield (three on the outer shield, two each on the middle and inner shields) at different locations and each shield's temperature was independently controlled. For example, for the inner shield a thermistor mounted on one end of the shield was used for providing temperature-feedback for the PI control, while another thermistor on the opposite end was used to monitor the controlled temperature. The monitored temperature of the inner shield was found to be stable to within $\pm 15 \mu\text{K}$ as stated in the manuscript. The corresponding data is now added as Supplementary Figure 6. We agree with the reviewer that stability of the reference temperature is of utmost importance and note that achieving such a stability of the inner shield is what made possible the sub-nW resolution claimed in our study.

***What is the meaning of the weak heat links between the shields?**

Thermal isolation of inner shield from the ambient temperature fluctuations is important to achieve the stability presented in this study. To achieve this isolation, the thermal conductance of the mechanical contacts between each shield needs to be minimized, while maintaining mechanical stability. This thermal isolation from the surroundings is what we call a weak thermal link. This has been further explained in the Methods section (High stability temperature system), quantifying the expected thermal conductance for the mechanical contacts used in our system

*** How was the thermal pulse input in Fig.2?**

To fully understand how this thermal pulse was applied we refer the referee to a detailed description of this technique in the Methods section titled "Thermal conductance measurement and power resolution". In brief, a DC voltage ($\sim 1.6 \text{ mV}$) applied on the thermistor embedded in the capillary tube, results in a power dissipation of $\sim 270 \text{ pW}$, while a simultaneous AC excitation facilitates temperature readout from the same thermistor. This DC voltage is varied in steps corresponding to a power of 270 pW as shown in Fig. 2a, while simultaneously recording the measured temperature changes.

Reviewers' Comments:

Reviewer #2:

Remarks to the Author:

The authors have addressed most issues appropriately in their rebuttal letter and changed the original manuscript accordingly. However, in their (good) response to my question on the small difference between SMR and AMR in *C. elegans*, the authors did not include this issue in the manuscript. It is important not to ignore 'conflicting data', but discuss it shortly in the manuscript. The authors should add one or two sentences on this (similar to what they responded in the rebuttal letter).

Response to reviewer's comments:

Reviewer #2 (Remarks to the Author):

The authors have addressed most issues appropriately in their rebuttal letter and changed the original manuscript accordingly. However, in their (good) response to my question on the small difference between SMR and AMR in *C. elegans*, the authors did not include this issue in the manuscript. It is important not to ignore 'conflicting data', but discuss it shortly in the manuscript. The authors should add one or two sentences on this (similar to what they responded in the rebuttal letter).

We thank the reviewer for the feedback. As suggested, we have now added a short description of the conflicting data in the "Metabolic measurements on N2 wild-type at different stages" subsection of the "Results" section, which reads as follows:

"We note that a previous study²⁸ reported a 50% difference in AMR and SMR as opposed to the 7% observed in our study, but a direct comparison of the results is challenging due to important differences in the experimental conditions. Specifically, we acquired the SMR of worms without restraining their activity while monitoring the metabolic rate whereas the earlier report²⁸ immobilized *C. elegans* either by levamisole-induced paralysis or by genetic paralysis."